# Diagnostic Performance of Risk of Malignancy Algorithm (ROMA), Risk of Malignancy Index (RMI) and Expert Ultrasound Assessment in a Pelvic Mass Classified as Inconclusive by International Ovarian Tumour Analysis (IOTA) Simple Rules

**DOI:** 10.3390/cancers14030810

**Published:** 2022-02-05

**Authors:** Siew Fei Ngu, Yu Ka Chai, Ka Man Choi, Tsin Wah Leung, Justin Li, Gladys S. T. Kwok, Mandy M. Y. Chu, Ka Yu Tse, Vincent Y. T. Cheung, Hextan Y. S. Ngan, Karen K. L. Chan

**Affiliations:** 1Department of Obstetrics and Gynaecology, The University of Hong Kong, Queen Mary Hospital, Hong Kong, China; ngusiewf@hku.hk (S.F.N.); gstkwok@hku.hk (G.S.T.K.); chumy@hku.hk (M.M.Y.C.); tseky@hku.hk (K.Y.T.); vytc@hku.hk (V.Y.T.C.); hysngan@hku.hk (H.Y.S.N.); 2Department of Obstetrics and Gynaecology, United Christian Hospital, Hong Kong, China; cyk095a@ha.org.hk; 3Department of Obstetrics and Gynaecology, Tseung Kwan O Hospital, Hong Kong, China; choikm@ha.org.hk; 4Department of Obstetrics and Gynaecology, Kwong Wah Hospital, Hong Kong, China; leungtw2@ha.org.hk; 5Li Ka Shing Faculty of Medicine, The University of Hong Kong, Hong Kong, China; just.li.328@gmail.com

**Keywords:** pelvic mass, ovarian cancer, biomarkers, HE4, CA125, risk of malignancy index (RMI), risk of malignancy algorithm (ROMA), international ovarian tumour analysis simple rules (IOTA)

## Abstract

**Simple Summary:**

The accurate prediction of malignancy for a pelvic mass detected on ultrasound allows for appropriate referral to specialised care. IOTA simple rules are one of the best methods but are inconclusive in 25% of cases, where subjective assessment by an expert sonographer is recommended but may not always be available. In the present paper, we evaluate the methods for assessing the nature of a pelvic mass, including IOTA with subjective assessment by expert ultrasound, RMI and ROMA. In particular, we investigate whether ROMA can replace expert ultrasound when IOTA is inconclusive. When IOTA was inconclusive, we found that expert ultrasound was more sensitive in diagnosing a malignant mass than ROMA, with no significant difference in the specificity or accuracy. All the assessment methods involving IOTA had similar accuracies, and they were more accurate than RMI or ROMA alone. Thus, IOTA should be the first step for assessing a pelvic mass. If inconclusive, an assessment by expert ultrasound is preferrable.

**Abstract:**

The accurate prediction of malignancy for a pelvic mass detected on ultrasound allows for appropriate referral to specialised care. IOTA simple rules are one of the best methods but are inconclusive in 25% of cases, where subjective assessment by an expert sonographer is recommended but may not always be available. In the present paper, we evaluate the methods for assessing the nature of a pelvic mass, including IOTA with subjective assessment by expert ultrasound, RMI and ROMA. In particular, we investigate whether ROMA can replace expert ultrasound when IOTA is inconclusive. This prospective study involves one cancer centre and three general units. Women scheduled for an operation for a pelvic mass underwent a pelvic ultrasound pre-operatively. The final histology was obtained from the operative sample. The sensitivity, specificity and accuracy for each method were compared with the McNemar test. Of the 690 women included in the study, 171 (25%) had an inconclusive IOTA. In this group, expert ultrasound was more sensitive in diagnosing a malignant mass compared to ROMA (81% vs. 63%, *p* = 0.009) with no significant difference in the specificity or accuracy. All assessment methods involving IOTA had similar accuracies and were more accurate than RMI or ROMA alone. In conclusion, when IOTA was inconclusive, assessment by expert ultrasound was more sensitive than ROMA, with similar specificity.

## 1. Introduction

Pelvic ultrasound is one of the most common investigations carried out for various gynaecological symptoms. Often, an ovarian or pelvic mass is found on ultrasound examination. The key question is whether the mass is benign or malignant. This has important implications because it determines the subsequent management. If the mass is likely to be benign, it can be observed or operated by general gynaecologists, usually laparoscopically. However, if it is malignant, the management would be very different. The patient would need to be referred to a gynaecological oncologist, as it has been shown that ovarian cancers managed by gynaecological oncologists would have better outcomes [1,2]. Therefore, the accurate determination of the likelihood of malignancy before the operation is important to avoid delaying the management of ovarian cancer, leading to a poorer prognosis. On the other hand, there would be unnecessary interventions and operative procedures with associated additional risks if a benign mass is managed as if it were malignant. Unlike other malignancies for which a biopsy can be taken to determine the nature of a lesion, a biopsy cannot be taken from an ovarian cyst as it would rupture the cyst, with the spillage of cyst contents into the abdomen, potentially leading to acute abdomen and up-staging of the cancer if the cyst turns out to be malignant. Therefore, methods for predicting the nature of the mass are needed.

Different methods have been developed to predict the likelihood of malignancy in a pelvic mass found on ultrasound [3,4,5,6]. One of the commonly used methods is the Risk of Malignancy Index (RMI), which is a product of the ultrasound morphological features, menopausal status and serum tumour marker CA125 [7,8,9]. This has been recommended by the Royal College of Obstetricians and Gynaecologists (U.K.) guidelines [10]. RMI has a pooled sensitivity of 78% (95% confidence interval, CI 71–85%) and specificity of 87% (95% CI 83–91%) [11].

The Risk of Malignancy Algorithm (ROMA) was developed using two tumour markers: CA125 and human epididymis protein 4 (HE4). CA125 is the most widely used tumour marker for ovarian cancer. However, CA125 is also raised in many conditions common in pre-menopausal women, such as fibroids, endometriosis and pelvic infection, and it is only raised in 50% of early-stage ovarian cancer [12]. Among other markers, HE4 has been the most promising [13]. HE4 was found to be elevated in more than half of ovarian tumours that do not express CA125 [14]. Therefore, a dual marker algorithm combining HE4 and CA125 was developed [15,16]. ROMA had similar sensitivity to CA125, but improved specificity, especially in pre-menopausal women. In addition, ROMA had an 83% accuracy in diagnosing early-stage disease [17]. In a meta-analysis comparing HE4, CA125 and ROMA, HE4 was found to be more useful in pre-menopausal women, while CA125 and ROMA were better in postmenopausal women [18]. The idea of combining ultrasound features and ROMA or HE4 was also investigated [19]. Combining ultrasound with HE4 can improve the sensitivity for detecting ovarian cancer compared to other algorithms. However, this has the same disadvantage in that it requires detailed ultrasound features, which are subject to variable interpretation, and therefore preclude its use in common clinical practice.

The International Ovarian Tumour Analysis (IOTA) study group developed a set of simple ultrasound-based rules [20,21] with a reported sensitivity of 92% and specificity of 96% [22,23,24,25,26]. Patients whose masses could not be classified with these rules would be referred for a subjective expert ultrasound assessment. With a combination of simple rules triage, followed by subjective expert ultrasound in those with inconclusive results, a sensitivity of 91–93% and specificity of 93% could be achieved. In a recent meta-analysis of 19 studies, IOTA simple rules were superior to all other methods, with an overall sensitivity of 93% and specificity of 81% [27].

IOTA simple rules seem to be a good option, but the results are inconclusive in 25% of the patients [20]. In this 25% of patients, an expert ultrasound is necessary. Unfortunately, the expertise of experienced ultrasound examiners in pelvic mass assessment is not easily transferred to less experienced examiners. Such expertise may not always be available. Therefore, we should have a strategy as an alternative if an expert is unavailable. With HE4 being more promising than CA125, especially in pre-menopausal women and early-stage disease, we evaluated the diagnostic performance of ROMA combined with IOTA simple rules. This study aims to assess whether ROMA could replace the need for assessment by experts for those women with inconclusive results from the IOTA simple rules.

## 2. Materials and Methods

### 2.1. Study Design

This was a multicentre prospective cohort study involving women scheduled for operation for a pelvic mass at the Department of Obstetrics and Gynaecology from one cancer centre (Queen Mary Hospital) and three general hospitals (United Christian Hospital, Pamela Youde Nethersole Hospital and Kwong Wah Hospital). The study was approved by the institutional review boards at each site and was registered at the HKU Clinical Trials Registry.

The primary research question was: “Among women with an adnexal mass scheduled for surgery who have inconclusive IOTA results, could ROMA predict malignancy better than assessment by expert ultrasound?”. We hypothesised that, in this group of women, the additional use of ROMA would achieve similar accuracy of prediction compared to referring these women for expert ultrasound assessments.

The secondary research questions were: “1. Among women with an adnexal mass scheduled for surgery who have conclusive IOTA results, which method (IOTA, ROMA or RMI) was the best at predicting ovarian malignancy?”; “2. Which method was the best at predicting ovarian malignancy among all women with ovarian pathology?” We compared the accuracy of IOTA simple rules followed by an expert ultrasound or ROMA, or RMI with ROMA alone, or RMI alone; “3. Does the performance of these tests vary by menopausal status (pre vs. postmenopausal), hospital settings (cancer specialist centre vs. general hospitals) or tumour histology or stage?”

Women over the age of 18 who were found to have a pelvic mass on ultrasound, magnetic resonance imaging, computed tomography or positron emission tomography scan and were scheduled for operation, including ovarian cystectomy and oophorectomy (laparoscopic or open), were eligible for the study and were recruited consecutively. The women who declined a transvaginal ultrasound scan, were pregnant, with a previous history of ovarian, peritoneal or fallopian tube cancer or unknown malignancy, and history of bilateral oophorectomy were excluded. Those women for whom surgical removal was delayed for more than 120 days from the date of the ultrasound examination were also excluded. All the participants were informed of the study and provided written consent.

The participants underwent an ultrasound assessment pre-operatively by gynaecologists with basic ultrasound training, who were not experts in ultrasounds. A small group of gynaecologists at each hospital participated in the scans for this study. Each scan was performed by one gynaecologist and he/she classified the patients into high risk or low risk for malignancy according to the IOTA simple rules and RMI (Table 1). Ultrasound machines with transvaginal probe frequencies ranging from 5–12 MHz were used. Greyscale and colour Doppler images were obtained for the morphology and blood flow. A transabdominal ultrasound was used for large masses that could not be completely visualised by transvaginal scan. The dominant mass was selected for analysis in women with multiple masses. During the ultrasound assessment, information on the following variables was collected, as suggested by the original IOTA papers [10,20]. The IOTA simple rules are based on five ultrasound features of malignancy (M-features) and five ultrasound features of benign lesion (B-features). The M-features are irregular solid tumour, presence of ascites, at least four papillary structures, irregular multilocular solid tumour ≥100 mm in diameter and very strong blood flow (IOTA colour score 4). The B-features are unilocular cysts, the presence of solid components of which the largest solid component <7 mm in diameter, acoustic shadows, smooth multilocular tumour <100 mm in diameter and no blood flow (IOTA colour score 1). The mass was classified as malignant if one or more M-features were present in the absence of a B-feature. The mass was classified as benign if one or more B-features were present in the absence of an M-feature. If both M-features and B-features were present or none of the features were present, the simple rules would be inconclusive. Those with inconclusive IOTA results underwent an expert ultrasound assessment by an expert sonographer or a gynaecologist who had obtained accreditation in Obstetric and Gynaecological Ultrasonography (a local competency-based assessment) and had been in practice in gynaecological ultrasound for more than five years. Ultrasound features required for RMI assessment, such as multilocular cysts, solid areas, metastases, ascites and bilateral lesions, were also recorded. The ROMA and RMI were calculated according to the suggestion by Moore and Jacobs, respectively [7,15]. The summary of the assessment methods used in predicting malignancy in a pelvic mass found on ultrasound investigated in this study is shown in Table 1.

Blood was taken for the CA125 and HE4 levels pre-operatively. A total of 10 mL of blood was collected into a serum or serum separator tube, centrifuged, aliquoted and stored at −20 °C or colder within 4 h. Blood samples taken from the three general hospitals were transported on dry ice to the cancer centre on the same day for central analysis. Samples were tested using the ARCHITECT CA125 II assay and ARCHITECT HE4 assay (Abbott Diagnostics, Abbott Park, IL, USA), according to the manufacturer’s instructions.

Surgery was performed by laparotomy or laparoscopy based on the surgeon’s decision. Excised tumour tissues were histologically examined at the local hospital. The histological assessment was performed by the pathologist without the knowledge of the ultrasound results.

For each woman, five prediction parameters, including (1) IOTA with expert ultrasound, (2) IOTA with ROMA, (3) IOTA with RMI, (4) ROMA alone and (5) RMI alone were calculated, and the results were correlated with the histology results: benign or malignant (including borderline tumours) from the surgery.

### 2.2. Sample Size Calculation

Our study focused on the group of women with inconclusive results from IOTA simple rules. Previous studies showed that this group represented about 25% of women undergoing IOTA assessment [20]. We assumed that the accuracy rate for ROMA was around 85%, and the actual difference between the 2 methods was 5% [17,20]. The range of non-inferiority was assumed at 5%. A sample size of 160 women would achieve 90% power to show non-inferiority between the 2 correlated accuracy rates. The calculation used a one-sided non-inferiority test of two correlated proportions. To achieve 160 women, a minimum of 640 women undergoing the operation would be needed.

### 2.3. Statistical Analysis

Data were analysed with the help of the expertise in the Biostatistics and Clinical Research Methodology Unit at the School of Public Health at the University of Hong Kong. The primary analysis focused on evaluating the sensitivity, specificity and accuracy of ROMA and assessment by expert ultrasound among women with inconclusive IOTA results. Sensitivity, specificity and accuracy (defined as the number of correct assessments divided by the number of all assessments) were compared using the McNemar test. Similar analyses were repeated for the secondary outcomes. Subgroup analyses were performed for different menopausal status, hospitals and histological subtypes. The 95% confidence interval of sensitivity, specificity and accuracy was calculated. The analyses were conducted using R version 4.1.0 (R Foundation for Statistical Computing Platform), IBM SPSS Statistics Version 25 or the link: https://www2.ccrb.cuhk.edu.hk/stat/confidence%20interval/McNemar%20Test.htm (accessed on 31 August 2021). *p* < 0.05 is considered statistically significant.

## 3. Results

### 3.1. Study Population

Between 30 April 2018 and 10 August 2020, 814 women were recruited: 408 were from a cancer centre, and 406 were from the general hospitals. A total of 678 women completed all study procedures, including the ultrasound, blood tests for CA125 and HE4 and surgery for the pelvic mass. An additional 19 women did not undergo surgery, but the diagnosis of malignancy was made on biopsy or cytology. Overall, 690 women had a histological or cytological diagnosis, and they were included in the final analysis (Figure 1). The median age was 46 years (range 18–69). The proportion of malignant pelvic masses was higher at the cancer centre than in the general hospitals (32.8% vs. 8.6%). The background demographic, histology and stage of the disease for the whole study population and those with inconclusive IOTA results are shown in Table 2a,b.

### 3.2. Amongst Women with an Inconclusive Result from the IOTA Simple Rules

We compared the accuracy of the two methods, the expert ultrasound versus ROMA in the group of women who had inconclusive IOTA results (n = 171, 25%) (Table 3). Expert ultrasound was more sensitive in diagnosing a malignant mass than ROMA (81% vs. 63%, *p* = 0.009), with no significant difference in the specificity and accuracy. For the IOTA with expert ultrasound, 15 out of 79 malignant tumours (19%) were wrongly predicted to be benign, while 29 (37%) were incorrectly predicted to be benign by IOTA with ROMA (*p* = 0.009). On the other hand, IOTA with expert ultrasound and IOTA with ROMA wrongly classified similar proportions of patients as malignant when the final pathology turned out to be benign (28% vs. 27%), respectively.

### 3.3. Amongst Women with a Conclusive Result from the IOTA Simple Rules

Of the 690 women with a histological or cytological diagnosis, the IOTA simple rules were conclusive in 519 women (75%). Amongst these women, IOTA and ROMA had similar sensitivities (81% and 82%, respectively), and both had better sensitivities than RMI (71%, *p* = 0.019 and *p* = 0.006, respectively). However, RMI had a better specificity than ROMA (94% vs. 85%, *p* < 0.001). Overall, IOTA was more accurate than ROMA or RMI in diagnosing a malignant mass (94% vs. 84% and 89%, *p* ≤ 0.001). Therefore, among all the women with conclusive IOTA results, IOTA is better at differentiating the malignant and benign cases correctly compared to ROMA or RMI (Table 4).

### 3.4. Amongst the Whole Population of Women with an Ovarian Pathology

We then explored which method best predicted ovarian malignancy for the whole study population (Table 5). Out of 640 women with an ovarian pathology, the IOTA with expert ultrasound was more sensitive than IOTA with ROMA or RMI (*p* = 0.015 and *p* = 0.001, respectively). ROMA alone was more sensitive than RMI alone (*p* = 0.007). The addition of IOTA to ROMA did not improve the sensitivity. All methods involving IOTA had similar specificities. RMI alone was more specific than ROMA alone (*p* < 0.001). The addition of IOTA to RMI further improved the specificity of RMI from 91% to 94% (*p* = 0.003).

In terms of accuracy, both IOTA with ROMA and IOTA with RMI were similar to IOTA with expert ultrasound. RMI alone was more accurate than ROMA alone (*p* = 0.054). The combination of IOTA with expert ultrasound, ROMA or RMI were all more accurate than ROMA or RMI alone (IOTA with expert vs. ROMA alone 89% vs. 82%, *p* < 0.001; IOTA with expert vs. RMI alone 89% vs. 84%, *p* = 0.001; IOTA with ROMA vs. ROMA alone 88% vs. 82%, *p* < 0.001; IOTA with ROMA vs. RMI alone 88% vs. 84%, *p* = 0.004; IOTA with RMI vs. ROMA alone 88% vs. 82%, *p* < 0.001; and IOTA with RMI vs. RMI alone 88% vs. 84%, *p* < 0.001).

### 3.5. Performance in Pre- and Postmenopausal Women

In pre-menopausal women, similar to the overall population, IOTA with expert ultrasound is more sensitive than IOTA with ROMA (81% vs. 73%, *p* = 0.035) (Table 6). ROMA alone was more sensitive than RMI alone (76% vs. 66%, *p* = 0.013). Adding IOTA to ROMA or RMI did not improve the sensitivity of either test alone. In terms of specificity, IOTA with expert ultrasound was similar to IOTA with ROMA (94% vs. 94%). RMI alone was more specific than ROMA alone (92% vs. 84%, *p* < 0.005). The addition of IOTA improved the specificity of RMI or ROMA alone (*p* < 0.005 for both). All strategies with IOTA had similar accuracies (IOTA with expert ultrasound vs. IOTA with RMI, *p* = 0.500; IOTA with expert ultrasound vs. IOTA with ROMA, *p* = 0.196; and IOTA with ROMA vs. IOTA with RMI, *p* = 0.166) and were more accurate than either ROMA or RMI alone (ROMA vs. IOTA with expert ultrasound, *p* = 0.00003; ROMA vs. IOTA with ROMA, *p* = 0.00001; ROMA vs. IOTA with RMI, *p* = 0.00001; RMI vs. IOTA with expert ultrasound, *p* = 0.011; RMI vs. IOTA with ROMA, *p* = 0.040; and RMI vs. IOTA with RMI, *p* = 0.001). RMI was more accurate than ROMA (*p* = 0.010).

In postmenopausal women, all IOTA strategies had similar sensitivities (Table 6). Adding IOTA to either ROMA or RMI did not improve the sensitivity, but IOTA with expert ultrasound was more sensitive than RMI alone (79% vs. 67%, *p* = 0.008). ROMA and RMI had similar sensitivities (72% vs. 66%, *p* = 0.150). All strategies involving IOTA had similar specificities. Similar to pre-menopausal women, all strategies with IOTA had similar accuracies. IOTA with ROMA was similar to ROMA alone (83% vs. 80%, *p* > 0.05), but more accurate than RMI alone (83% vs. 77%, *p* = 0.027). ROMA and RMI alone had similar accuracies (80% vs. 77%).

IOTA with expert and IOTA with ROMA had similar sensitivities in pre- and postmenopausal women (81% vs. 79%, *p* > 0.05), but both were more accurate in premenopausal women (92% vs. 84%, *p* = 0.009; 90% vs. 83%, *p* = 0.017, respectively). ROMA and RMI had similar sensitivities and specificities in pre- and postmenopausal women, but RMI was more accurate in pre-menopausal women than in postmenopausal women (87% vs. 77%, *p* = 0.003).

### 3.6. Performance in a Cancer Centre vs. General Hospitals (for Patients with an Ovarian Pathology)

In the cancer centre (n = 315), IOTA with expert ultrasound is more sensitive, but less specific than IOTA with ROMA (83% vs. 76%, *p* = 0.033 and 87% vs. 91%, *p* = 0.040, respectively), with overall no difference in the accuracy (85 % vs. 85%) (Table 7). ROMA and RMI had similar sensitivities, but RMI was more specific (87% vs. 80%) with an overall similar accuracy. In the general units (n = 325), there was no significant difference between IOTA with expert ultrasound and IOTA with ROMA in sensitivity, specificity and accuracy. IOTA with expert or ROMA, or ROMA alone, was more sensitive than RMI alone. The addition of IOTA to RMI improved the sensitivity of RMI alone (60% vs. 42%, *p* = 0.008) and approximated that to ROMA alone. In both types of hospitals, the addition of IOTA improved the accuracy of ROMA or RMI alone.

### 3.7. Sensitivity in Diagnosing Early-Stage (Stage 1) Cancer

There were 51 stage 1 cancers out of 142 ovarian cancers. The histology included 19 clear cell, 12 serous, 11 endometrioid, 6 mucinous, 1 seromucinous and 2 mixed clear cell and endometrioid. The sensitivity in diagnosing early-stage cancer for the different strategies is shown in Table 8. IOTA with expert ultrasound had a similar sensitivity compared with IOTA with ROMA/RMI or ROMA alone. All strategies involving IOTA were more sensitive than RMI alone (IOTA with expert 81% vs. RMI alone 58%, *p* = 0.003; IOTA with ROMA 72% vs. RMI alone 58%, *p* = 0.031; and IOTA with RMI 70% vs. RMI alone 58%, *p* = 0.035). ROMA alone was more sensitive than RMI alone (70% vs. 58%), but this did not reach statistical significance (*p* = 0.061). IOTA with ROMA was not more accurate than IOTA with RMI (Table 8).

### 3.8. Performance for Different Histological Types of Ovarian Cancer

There were 36 borderline tumours included in the study. All strategies investigated in this study had a poorer sensitivity in diagnosing borderline tumours ranging from 36% to 57%. There was no significant difference in the various methods. We also tried to explore the sensitivities for all non-epithelial tumours (n = 28). Again, the overall sensitivity was slightly lower than for epithelial tumours (57–71% vs. 67–80%), but they did not reach statistical significance. This analysis was limited by the relative rarity of non-epithelial tumours.

## 4. Discussion

Many strategies and ultrasound criteria were investigated to predict malignancy in a pelvic mass detected on ultrasound. IOTA simple rules, ROMA and RMI were amongst the most commonly adopted methods, clinically. In this study, we compared the performances of these common strategies, particularly those for which the IOTA simple rules were inconclusive. The percentage of malignancy in our study (21%) was in the same range as other published series (Table 9). Likewise, the percentage of malignancy among women with inconclusive IOTA in our study (31%) was also comparable to other reported series (40% in Timmerman et al.) [20]. Similar to previous findings, IOTA, when yielding a conclusive result, was more accurate than ROMA and RMI. However, we found that IOTA was inconclusive in 25% of the women, similar to the percentage reported in the literature [20]. Subjective assessment by an expert sonographer was recommended in these cases with inconclusive IOTA results [20]. Nonetheless, there was limited data on how to assess this group of women if an expert was not available. Timmerman et al. reported the sensitivity of RMI or logistic regression in these cases, and they found that these were inferior to subjective assessment by an expert. The sensitivity reported was only 50% for RMI, while for subjective assessment it was 89% [20]. This was similar to our findings. The usefulness of ROMA in this situation is lacking. Potentially, with an additional tumour marker (HE4), it can out-perform RMI. It was reported that the addition of ROMA to expert subjective assessment did not further improve the diagnostic accuracy [28], but whether ROMA can replace subjective expert assessment was not reported. In this study, we explored the performance of ROMA in this group of patients. We found that ROMA was less sensitive than expert ultrasound. This suggested that expert ultrasound should be the investigation of choice when IOTA yielded inconclusive results. However, ROMA and RMI are possible options if expertise is not available since they have similar accuracy to expert ultrasound, despite a lower sensitivity. Since ROMA involves measuring an additional tumour marker, further considerations on the cost-effectiveness should be explored.

For the whole study population (IOTA conclusive and inconclusive), we investigated which strategy would be best at predicting ovarian cancer. Again, we found that IOTA with expert ultrasound was the most sensitive. All three strategies involving IOTA had similar accuracies and were more accurate than just ROMA or RMI alone. This suggested that we should adopt a strategy that involved IOTA as the first step in the assessment. If IOTA was inconclusive, we could use either expert assessment, ROMA or RMI to further predict the nature of the mass. We further explored if this applied to both cancer centres and general units where the prevalence of malignancy was very different. We found that the sensitivity of all the strategies was higher in the cancer centre, which was likely the result of a higher prevalence of malignancy in the cancer centre. Although a test’s sensitivity is not expected to vary with disease prevalence, the fact that different sensitivities were observed suggested that the test may not function in the same way in the cancer centre and general units. The implications of a false positive or negative test would also be different in the cancer centre compared to the general units. In the general units, a missed diagnosis of malignancy (i.e., a low sensitivity) would lead to the need for a second comprehensive operation for the definitive cancer management after the simple surgery was performed at the general units. Meanwhile, an over-diagnosis of malignancy (i.e., low specificity) would lead to unnecessary referrals to the cancer centre for primary surgery. From the patient’s view, it would be more important for the test to have a high sensitivity to avoid a second operation. Our study found that both IOTA with expert ultrasound and IOTA with ROMA offered the best sensitivity in the general units. However, we realised that expertise in ultrasound or ROMA might not be readily available at the general units. In this situation, adding IOTA to RMI (i.e., perform IOTA first, if inconclusive, use RMI) would be better than RMI alone since this would improve the sensitivity from 42% to 60%.

In a cancer centre, a missed diagnosis of malignancy might not lead to the need for a second operation because expertise would be available at the hospital. The gynae-oncologist could be called into the theatre to complete the full cancer operation when the frozen section of the mass showed malignancy, even though this would not be ideal logistically. On the other hand, if a benign mass was wrongly predicted to be malignant pre-operatively (i.e., low specificity), the woman may have undergone an unnecessary laparotomy with a midline incision rather than a minimally invasive procedure. In our centre, laparotomy is used in most patients with a suspected ovarian malignancy, although laparoscopic surgery may be considered in selected patient with isolated adnexal mass. From the patient’s point of view, one might argue that a test with a high specificity would be more important in the cancer centre. Our results show that IOTA with ROMA is the most specific, and this might be the test of choice in a cancer centre. In situations where ROMA was unavailable, IOTA with expert ultrasound would be the second choice, as it was more specific than RMI alone.

We also investigated any differences in the performances of the various strategies in pre- and postmenopausal women. We found that IOTA strategies appeared to perform better in pre-menopausal women. This agrees with the subgroup analysis of a meta-analysis, which showed a higher accuracy of IOTA simple rules in pre-menopausal women, possibly due to better diagnosis of endometriotic or dermoid cysts [27]. Amongst the IOTA strategies, Timmerman et al. found that expert ultrasound was more sensitive than RMI in those with inconclusive results in both pre- and postmenopausal women [20]. Our study also had similar findings. In addition, we found that IOTA with expert ultrasound was also superior to IOTA with ROMA in pre-menopausal women, but not in postmenopausal women. Similar to the other reported series [33,34], we found that RMI was more accurate in pre-menopausal women than in postmenopausal women, whereas ROMA had a similar accuracy in both pre and postmenopausal women.

We noted that RMI alone had a particularly low sensitivity for detecting early disease. This was expected since CA125, the tumour marker included in the calculation of RMI, is well known to be elevated in late but not early disease. Adding IOTA to RMI would improve the sensitivity from 58% to 70%. The IOTA algorithm may have a role in ovarian cancer screening.

### Strengths and Weaknesses

A strength of this study was its prospective multicentre design, involving both cancer centre and general units. This allows the results to be applicable in various settings with different prevalences of malignancy. Another strength was the availability of the reference standard (histological diagnosis) in nearly all the cases. In this study, assessments by IOTA were conducted by gynaecologists who were not ultrasound experts. This reflects the original purpose of IOTA for triaging patients in the general population for referral to specialised ultrasound experts. Our study results, therefore, can reflect realistic clinical scenarios. On the other hand, our study included patients who were already scheduled for surgery, and thus represented a selected group of women. In clinical practice, apart from deciding for the need for referral to the gynaecological oncologist, another dilemma would be whether a patient needs a surgery or not. Thus, our results on these different strategies may not be useful in informing the decision for surgery. Another limitation was that the gynaecologists conducting the IOTA assessment in this study did not attend any specific training courses in using IOTA. This may have affected the accuracy of IOTA and can explain the lower sensitivity in our population compared to those reported in the literature. Another limitation of the study was the different definition of the “expert” performing the expert ultrasound assessment in this study, compared to published training requirements. Internationally published training requirements for an expert ultrasound examiner specified that the expert should spend most of the time performing ultrasound examinations [35]. In our study, the experts were defined as gynaecologists with more than five years of ultrasound experience. These different definitions may affect the overall accuracy of the diagnostic tests in this study. Nonetheless, despite the different definitions of “experts”, the superiority of expert ultrasound was still demonstrated in this study.

Similar to the other studies, our study evaluated the diagnostic performance of some of the more commonly used methods to predict the risk of malignancy of an adnexal mass. However, these studies did not consider some relevant clinical factors that may affect management, many of which would make these strategies inappropriate from practical perspective. For example, if a woman has a computed tomography that is suspicious for carcinomatosis, her surgery will be performed by a gynaecologic oncologist. Hence, the results of the IOTA, RMI or ROMA or any combination of the strategies are irrelevant. Similarly, CA125 was widely used and routinely performed in postmenopausal women with adnexal masses. A postmenopausal woman with a mass and a markedly elevated CA125 would have her surgery performed by a gynaecologic oncologist, regardless of the RMI, ROMA or IOTA results. Therefore, what is needed to actually assess the clinical value of these strategies is a prospective randomised study that evaluates their performance in patients who do not exhibit the obvious findings of malignancy, such as ascites, evidence of metastatic disease and elevated CA125, in which the management is dependent on the conclusion of the strategies. Then, outcomes, such as the need for second surgery, morbidity and overall survival, can be evaluated.

## 5. Conclusions

Apart from the strategies investigated in this study, many different methods or classification systems for predicting malignancy in pelvic masses detected on ultrasound were proposed, such as logistic regressions models [36] and, more recently, the simple rules risk model [37] and the ADNEX model [38]. There is increasing evidence supporting these new prediction models. Recently, ESGO/ISUOG/IOTA/ESGE published a consensus statement of pre-operative diagnosis of ovarian tumours. Subjective assessment by expert ultrasound has the best performance, but if such expertise is not available, the IOTA ADNEX model and IOTA simple rule risk model are recommended [39]. These methods require an online calculator or smartphone apps. The actual uptake of these newer assessment models into daily clinical practice is yet to be defined. Currently, the methods investigated in this study, namely IOTA simple rules, RMI and ROMA, were most widely used in clinical practice, mainly due to ease of use. Our study supported using IOTA simple rules as the first step in assessing a pelvic mass. For those with inconclusive results, an expert assessment would be superior to ROMA or RMI. If expertise is not available, assessment by ROMA or RMI are acceptable. The addition of IOTA to ROMA or RMI was better than ROMA or RMI alone. This would apply to both cancer centres and general units settings.

## Figures and Tables

**Figure 1 cancers-14-00810-f001:**
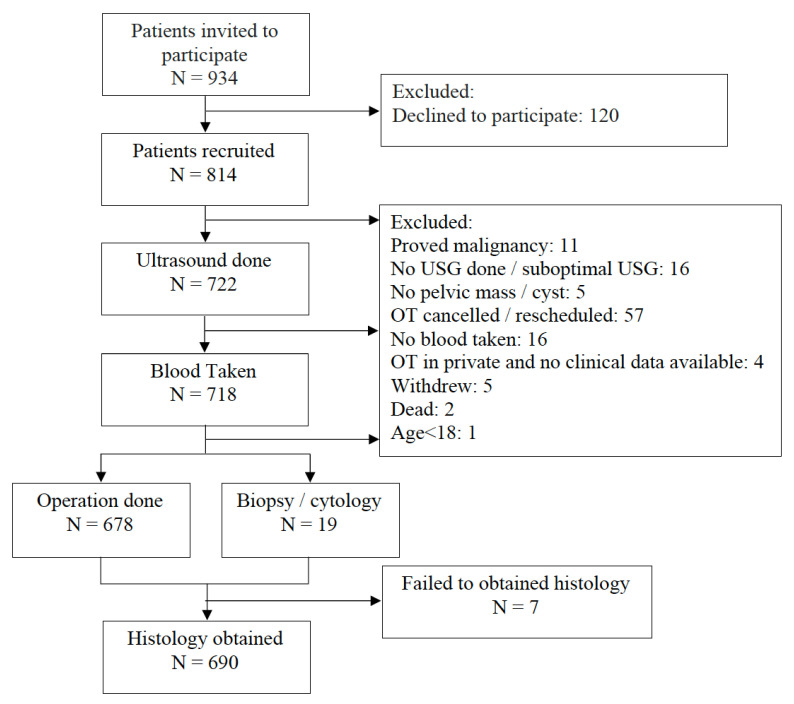
The flow of participants.

**Table 1 cancers-14-00810-t001:** Methods used in predicting malignancy in a pelvic mass found on ultrasound investigated in this study.

Methods	Components	Risk of Malignancy	
		High Risk	Low Risk
IOTA simple rules (IOTA)	Ultrasound assessment using 5 benign (B-features) and 5 malignant features (M-features)	Presence of >1 M-features and absence of B-features	Presence of >1 B-features and absence of M-features
Risk of malignancy algorithm (ROMA)	Calculation of risk by an algorithm taking into account the menopausal status, CA125 and HE4 levels Premenopausal Predictive Index (PI) = −12.0 + 2.38 × LN (natural log) [HE4] + 0.0626 × LN[CA125] Postmenopausal PI = −8.09 + 1.04 × LN[HE4] + 0.732 × LN[CA125] ROMA = exp(PI)/[1 + exp(PI)] × 100	Premenopausal: ROMA ≥ 7 .4 Postmenopausal: ROMA ≥ 25.3	Premenopausal: ROMA < 7.4 Postmenopausal: ROMA < 25.3
Risk of malignancy index (RMI)	Calculation of risk by ultrasound score (U), menopausal status (M) and CA125 level RMI = U × M × CA125	RMI ≥ 200	RMI < 200
Expert ultrasound	Subjective assessment by expert sonographer	Assessment suggestive of malignancy	Assessment suggestive of benign tumour

**Table 2 cancers-14-00810-t002:** (**a**) Demographics for the whole study population. (**b**) Demographics for women with inconclusive IOTA simple rules results.

**(a)**				
		**Cancer Centre**	**General Units**	**Total**
No. of patients		341 (49.4%) *	349 (50.6%) *	690 (100%) *
Age (median)		47 (18–85)	45 (19–89)	46 (18–89)
Menopausal status				
	Postmenopausal	113 (33.1%)	99 (28.4%)	212 (30.7%)
	Pre-menopausal	228 (66.9%)	250 (71.6%)	478 (69.3%)
No. of ovarian malignancies (%)		112 (32.8%)	30 (8.6%)	142 (20.6%)
Histology				
Ovarian				
	Benign	184 (54.0%)	275 (78.8%)	459 (66.5%)
	Endometriotic cyst	84 (45.7%)	97 (35.3%)	181 (39.4%)
	Dermoid	32 (17.4%)	71 25.8%)	103 (22.4%)
	Serous/mucinous cystadenoma	34 (18.5%)	57 (20.7%)	91 (19.8%)
	Fibroma	5 (2.7%)	7 (2.5%)	12 (2.6%)
	Functional cyst	6 (3.3%)	8 (2.9%)	14 (3.1%)
	Hydrosalpinx	1 (0.5%)	2 (0.7%)	3 (0.7%)
	Mixed	3 (1.6%)	1 (0.4%)	4 (0.9%)
	Others/unspecified	19 (10.3%)	32 (11.6%)	51 (11.1%)
	Malignant	112 (32.8%)	30 (8.6%)	142 (20.6%)
	Serous	28 (25.0%)	9 (30.0%)	37 (26.1%)
	Mucinous	5 (4.5%)	2 (6.7%)	7 (4.9%)
	Clear cell	25 (22.3%)	5 (16.7%)	30 (21.1%)
	Endometrioid	18 (16.1%)	7 (23.3%)	25 (17.6%)
	Mixed	13 (11.6%)	0 (0%)	13 (9.2%)
	Sex cord stromal/germ cell	4 (3.6%)	2 (6.7%)	6 (4.2%)
	Metastatic	10 (8.9%)	4 (13.3%)	14 (9.9%)
	Others	9 (8.0%)	1 (3.3%)	10 (7.0%)
	Borderline	16 (4.7%)	20 (5.7%)	36 (5.2%)
	Malignant/borderline	1 (0.3%)	0 (0%)	1 (0.1%)
Non-ovarian				
	Benign	10 (2.9%)	23 (6.6%)	33 (4.8%)
	Malignant	18 (5.3%)	1 (0.3%)	19 (2.8%)
FIGO staging				
	I	36 (39.6%)	15 (65.2%)	51 (44.7%)
	II	11 (12.1%)	4 (17.4%)	15 (13.2%)
	III	23 (25.3%)	2 (8.7%)	25 (21.9%)
	IV	10 (11.0%)	2 (8.7%)	12 (10.5%)
	Unstaged	11 (12.1%)	0 (0%)	11 (9.6%)
No. of inconclusive IOTA (%)		105 (30.8%)	66 (18.9%)	171 (24.8%)
(**b**)				
		**Cancer Centre**	**General Units**	**Total**
No. of patients		105 (61.4%) *	66 (38.6%) *	171 (100%) *
Age (median)		49 (21–84)	46.5 (22–83)	48 (21–84)
Menopausal status				
	Postmenopausal	46 (43.8%)	21 (31.8%)	67 (39.2%)
	Pre-menopausal	59 (56.2%)	45 (68.2%)	104 (60.8%)
No. of ovarian malignancies (%)		46 (43.8%)	7 (10.6%)	53 (31.0%)
Histology				
Ovarian				
	Benign	37 (35.2%)	48 (72.7%)	85 (49.7%)
	Endometriotic cyst	11 (29.7%)	15 (31.3%)	26 (30.6%)
	Dermoid	7 (18.9%)	15 (31.3%)	22 (25.9%)
	Serous/mucinous cystadenoma	12 (32.4%)	8 (16.7%)	20 (23.5%)
	Fibroma	2 (5.4%)	5 (10.4%)	7 (8.2%)
	Functional cyst	1 (2.7%)	0 (0%)	1 (1.2%)
	Others/unspecified	4 (10.8%)	5 (10.4%)	9 (10.6%)
	Malignant	46 (43.8%)	7 (10.6%)	53 (31.0%)
	Serous	8 (17.4%)	1 (1.5%)	9 (5.3%)
	Mucinous	3 (6.5%)	0 (0%)	3 (1.8%)
	Clear cell	16 (34.8%)	1 (1.5%)	17 (9.9%)
	Endometrioid	4 (8.7%)	2 (3.0%)	6 (3.5%)
	Mixed	4 (8.7%)	0 (0%)	4 (2.3%)
	Sex cord stromal/germ cell	2 (4.3%)	1 (1.5%)	3 (1.8%)
	Metastatic	6 (13.0%)	1 (1.5%)	7 (4.1%)
	Others	3 (6.5%)	1 (1.5%)	4 (2.3%)
	Borderline	9 (8.6%)	7 (10.6%)	16 (9.4%)
	Malignant/borderline	1 (1.0%)	0 (0%)	1 (0.6%)
Non-ovarian				
	Benign	4 (3.8%)	3 (4.5%)	7 (4.1%)
	Malignant	8 (7.6%)	1 (1.5%)	9 (5.3%)
FIGO staging				
	I	17 (43.6%)	3 (50.0%)	20 (44.4%)
	II	2 (5.1%)	2 (33.3%)	4 (8.9%)
	III	12 (30.8%)	1 (16.7%)	13 (28.9%)
	IV	2 (5.1%)	0 (0%)	2 (4.4%)
	Unstaged	6 (15.4%)	0 (0%)	6 (13.3%)

* Row percentages, all others are column percentages.

**Table 3 cancers-14-00810-t003:** (**a**) Correlation of the diagnostic tests results with the histology results in women with inconclusive IOTA results (n = 171). (**b**) Sensitivity, specificity and accuracy for expert ultrasound and ROMA in women with inconclusive IOTA results (n = 171).

**(a)**			
	**Risk of Malignancy**	**Histology**	
		**Malignant**	**Benign**
Expert ultrasound	High risk	64	26
	Low risk	15	66
ROMA	High risk	50	25
	Low risk	29	67
(**b**)			
	**Sensitivity** (**95% CI**)	**Specificity** (**95% CI**)	**Accuracy** (**95%CI**)
Expert ultrasound	81.0% (70.3–88.6%)	71.7% (61.2–80.4%)	76.0% (68.8–82.1%)
ROMA	63.3% (51.6–73.6%)	72.8% (62.4–81.3%)	68.4% (60.8–75.2%)

**Table 4 cancers-14-00810-t004:** (**a**) Correlation of the diagnostic tests with the histology results in women with conclusive IOTA results (n = 519). (**b**) Sensitivity, specificity and accuracy for IOTA, ROMA and RMI in women with conclusive IOTA results (n = 519).

**(a)**			
	**Risk of Malignancy**	**Histology**	
		**Malignant**	**Benign**
IOTA	High risk	96	10
	Low risk	23	390
ROMA	High risk	97	59
	Low risk	22	341
RMI	High risk	84	23
	Low risk	35	377
(**b**)			
	**Sensitivity (95% CI)**	**Specificity (95% CI)**	**Accuracy (95%CI)**
IOTA	80.7% (72.2–87.1%)	97.5% (95.3–98.7%)	93.6% (91.1–95.5%)
ROMA	81.5% (73.1–87.8%)	85.3% (81.3–88.5%)	84.4% (80.9–87.3%)
RMI	70.6% (61.4–78.4%)	94.3% (91.4–96.2%)	88.8% (85.7–91.3%)

**Table 5 cancers-14-00810-t005:** Diagnostic accuracy for five different strategies in all women with an ovarian pathology (n = 640).

	Sensitivity (95% CI)	Specificity (95% CI)	Accuracy (95%CI)
IOTA + expert	79.9% (73.1–85.3%)	92.8% (90.0–94.9%)	89.2% (86.5–91.5%)
IOTA + ROMA	73.2% (66.0–79.4%)	93.7% (91.0–95.7%)	88.0% (85.1–90.3%)
IOTA + RMI	72.1% (64.8–78.4%)	94.1% (91.5–96.0%)	88.0% (85.1–90.3%)
ROMA alone	74.3% (67.1–80.4%)	84.4% (80.7–87.5%)	81.6% (78.3–84.4%)
RMI alone	66.5% (59.0–73.2%)	91.1% (88.0–93.5%)	84.2% (81.1–86.9%)

**Table 6 cancers-14-00810-t006:** Sensitivity, specificity and accuracy for pre- and postmenopausal women.

	Sensitivity (95% CI)	Specificity (95% CI)	Accuracy (95%CI)
Premenopausal			
IOTA + expert	80.9% (70.9–88.2%)	94.1% (91.0–96.2%)	91.5% (88.4–93.8%)
IOTA + ROMA	73.0% (62.4–81.6%)	94.4% (91.3–96.4%)	90.1% (86.9–92.6%)
IOTA + RMI	71.9% (61.2–80.7%)	96.1% (93.3–97.7%)	91.2% (88.1–93.6%)
ROMA alone	76.4% (66.0–84.5%)	84.0% (79.7–87.6%)	82.5% (78.5–85.8%)
RMI alone	66.3% (55.4–75.8%)	92.4% (89.0–94.9%)	87.2% (83.6–90.1%)
Postmenopausal			
IOTA + expert	78.9% (68.8–86.5%)	88.6% (80.5–93.7%)	84.1% (78.0–88.8%)
IOTA + ROMA	73.3% (62.8–81.9%)	91.4% (83.9–95.8%)	83.1% (76.9–87.9%)
IOTA + RMI	72.2% (61.6–80.9%)	87.6% (79.4–93.0%)	80.5% (74.1–85.7%)
ROMA alone	72.2% (61.6–80.9%)	85.7% (77.2–91.5%)	79.5% (73.0–84.8%)
RMI alone	66.7% (55.9–76.0%)	86.7% (78.3–92.3%)	77.4% (70.8–83.0%)

**Table 7 cancers-14-00810-t007:** Diagnostic performance in the cancer unit vs. general units.

	Sensitivity (95% CI)		Specificity (95% CI)		Accuracy (95%CI)	
	Cancer	General	Cancer	General	Cancer	General
IOTA + expert	82.9% (75.1–88.8%)	72.0% (57.3–83.3%)	87.1% (81.2–91.4%)	96.7% (93.7–98.4%)	85.4% (80.9–89.0%)	92.9% (89.4–95.4%)
IOTA + ROMA	76.0% (67.5–82.9%)	66.0% (51.1–78.4%)	91.4% (86.2–94.8%)	95.3% (91.9–97.4%)	85.1% (80.5–88.7%)	90.8% (87.0–93.6%)
IOTA + RMI	76.7% (68.3–83.5%)	60.0% (45.2–73.3%)	91.9% (86.8–95.3%)	95.6% (92.3–97.6%)	85.7% (81.2–89.3%)	90.2% (86.3–93.1%)
ROMA alone	79.8% (71.7–86.2%)	60.0% (45.2–73.3%)	79.6% (72.9–85.0%)	87.6% (83.0–91.2%)	79.7% (74.7–83.9%)	83.4% (78.8–87.2%)
RMI alone	76.0% (67.5–82.9%)	42.0% (28.5–56.7%)	87.1% (81.2–91.4%)	93.8% (90.1–96.2%)	82.5% (77.8–86.5%)	85.8% (81.5–89.4%)

**Table 8 cancers-14-00810-t008:** Sensitivity for predicting early-stage (stage 1) ovarian cancer.

	Sensitivity
IOTA + expert	80.7% (67.7–89.5%)
IOTA + ROMA	71.9% (58.3–82.6%)
IOTA + RMI	70.2% (56.4–81.2%)
ROMA alone	70.2% (56.4–81.2%)
RMI alone	57.9% (44.1–70.6%)

**Table 9 cancers-14-00810-t009:** The percentage of malignancy, inconclusive IOTA, sensitivity and specificity in previous and present studies.

	Number of Patients	Malignancy Prevalence %	Inconclusive IOTA %	Sensitivity %	Specificity %
Timmerman D et al., 2010 [20]	997	28	23	90	93
Hartman CA et al., 2012 [29]	110	28	17	84	86
Alcazar JL et al., 2013 [30]	340	16	21	89	96
Sayasneh A et al., 2013 [26]	255	29	17	86	94
Nunes N et al., 2014 [23]	303	44	22	94	89
Ruiz de Gauna B et al., 2015 [25] centre A	114	27	18	100	89
Ruiz de Gauna B et al., 2015 [25] centre B	133	11	18	86	88
Knafel A et al., 2013 [31]	226	NA	18	95	74
Piovano E et al., 2017 [32]	391	21	11	82	92
Current study	640	21	25	80	93

## Data Availability

Not applicable.

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
