# Peer review of "Diagnostic Performance of Risk of Malignancy Algorithm (ROMA), Risk of Malignancy Index (RMI) and Expert Ultrasound Assessment in a Pelvic Mass Classified as Inconclusive by International Ovarian Tumour Analysis (IOTA) Simple Rules"

_cancers, 2022, doi:10.3390/cancers14030810_

Round 1

Reviewer 1 Report

I read your manuscript “Diagnostic Performance of Risk of Malignancy Algorithm (ROMA), Risk of Malignancy Index (RMI) and Expert Ultra-sound Assessment in a Pelvic Mass Classified as Inconclusive by International Ovarian Tumour Analysis (IOTA) Simple Rules” with great interest. First of all, I want to congratulate you for the clarity and quality of your study design and results. The study is very well done and can really contribute to the reader as you use the simple rules of IOTA without training given by the IOTA group, characterizing the effectiveness of this classification.

Introduction: your introduction is very clear and written in a very simple and objective way. However, although you opted for basic references, for example the first publications by Bast et al, 1983, or Jacobs et al. 1990, what I understand perfectly, most of the references in the introduction are very old, and in the last 5 years many studies have been carried out to validate RMI, IOTA simple rules or ROMA. Just to illustrate, below I put some recent references that I believe are well related to your study (and in none of them I have a personal interest or conflict of interest)

Van Calster B, Valentin L, Froyman W, Landolfo C, Ceusters J, Testa AC, Wynants L, Sladkevicius P, Van Holsbeke C, Domali E, Fruscio R, Epstein E, Franchi D, Kudla MJ, Chiappa V, Alcazar JL, Leone FPG, Buonomo F, Coccia ME, Guerriero S, Deo N, Jokubkiene L, Savelli L, Fischerová D, Czekierdowski A, Kaijser J, Coosemans A, Scambia G, Vergote I, Bourne T, Timmerman D. Validation of models to diagnose ovarian cancer in patients managed surgically or conservatively: multicentre cohort study. BMJ. 2020 Jul 30;370:m2614. doi: 10.1136/bmj.m2614.

Chacón E , Dasí J , Caballero C , et al . Risk of ovarian malignancy algorithm versus risk malignancy index-I for preoperative assessment of adnexal masses: a systematic review and meta-analysis. Gynecol Obstet Invest 2019;84:591–8.doi:10.1159/000501681

Khoiwal K , Bahadur A , Kumari R , et al . Assessment of diagnostic value of serum CA-125 and risk of malignancy index scoring in the evaluation of adnexal masses. J Midlife Health 2019;10:192–6.doi:10.4103/jmh.JMH_84_19

Zhang S , Yu S , Hou W , et al . Diagnostic extended usefulness of RMI: comparison of four risk of malignancy index in preoperative differentiation of borderline ovarian tumors and benign ovarian tumors. J Ovarian Res 2019;12:87. doi:10.1186/s13048-019-0568-3

Westwood M , Ramaekers B , Lang S , et al . Risk scores to guide referral decisions for people with suspected ovarian cancer in secondary care: a systematic review and cost-effectiveness analysis. Health Technol Assess 2018;22:1–264.doi:10.3310/hta22440

Meys EMJ , Kaijser J , Kruitwagen RFPM , et al . Subjective assessment versus ultrasound models to diagnose ovarian cancer: a systematic review and meta-analysis. Eur J Cancer 2016;58:17–29.doi:10.1016/j.ejca.2016.01.007

Materials and Methods

Excellent study design.

I think only figure 1 could have a review:  “Figure 1. Schematic drawing illustrates Strategy A vs Strategy B.” just to confirm that the picture is clear: you mean that all women with inconclusive IOTA simple rules classification results were referred to perform a US with an expert and that all of them also collected CA125 and HE4?  Where is the RMI in the figure? Later you say that all women do the US already with the classification of the simple rules and the RMI. For statistical analysis the techniques were compared?

I think it is important to define what is low risk and high risk for each variable, as it will be used in the results (example tables 3, 4, 5, 6 etc…)

Sample size calculation and Statistical analysis: OK

Result: clear and really well used to represent the different scenarios

Discussion: good and already using much more recent references

Reviewer 2 Report

The authors present an analysis of the performance of IOTA vs RMI vs ROMA or combinations of those algorithms in predicting ovarian cancer among women scheduled for surgery for an adnexal mass and either a cancer center or community hospital.

Overall, the paper is reasonably well written from a clarity of writing perspective, although it needs to be shortened significantly to be more concise. The authors present several different comparisons of the combinations of IOTA, RMI and ROMA in pre and postmenopausal women and in the cancer center vs community hospital.

There are many similar studies that evaluate these algorithms in their practice settings. However most other studies do not look at expert ultrasound review as an option so the authors should be commended for this. My main complaints with the paper are 1) failure to articulate a clear research question and have the analysis clearly address that one research question and 2) assumptions that are made implicit in their conclusions and in the discussion that are not necessarily true. These and other comments are detailed below:

  1. The research question should be more clearly stated. On the one hand, it seems to be something like “Among women with an adnexal mass scheduled for surgery who have inconclusive IOTA scores, does ROMA predict malignancy better than expert ultrasound evaluation?” However, they then analyze the entire cohort, not just the women with the inconclusive IOTA score so the population of interest--women with inconclusive IOTA scores-- gets lost in the subsequent analysis and discussion. The failure to articulate a clear research question in which the POPULATION subject to the research question is clearly defined (all women having surgery for an adnexal mass vs women with an inconclusive IOTA score) and the COMPARISON and OUTCOME are clearly defined is a major problem that makes the overall paper more difficult to understand.
  2. Line 70 states that “the most commonly used method” to predict likelihood of malignancy is RMI. I am not aware of evidence showing this to be true. Certainly in the US, RMI is rarely used. If the authors do not have data to substantiate the claim that RMI is the “most commonly used method” this statement should be removed.
  3. Line 88: refers to the IOTA simple rules as “complicated,” that being the problem with its use. I would disagree with this—the issue is not that ultrasound is complicated but rather that there is variable interpretation of ultrasound features.
  4. Line 92: Would suggest removing “To circumvent the issue of using complicated ultrasound features in the prediction models,” and simply start the next sentence with “The International…”
  5. In Materials and Methods/Study Design, the content should be rearranged to start with the 3rd paragraph which states the study design, then follow with the hypothesis.
  6. The time interval from which patients were recruited should be clearly stated (eg from X date to Y date). A flowchart of the cohort assembly would be useful. The authors claim that everyone was prospectively consented but it seems there were no patients excluded due refusal of consent? This seems unlikely in the real world. If this is the case, it calls into question the validity of the consent process. This should be clarified.
  7. Line 140 “They were classified into high risk or low risk for malignancy according to the IOTA simple rules and RMI.” The authors should describe exactly how this classification was done—was this done separately for IOTA and RMI and then combined? If so, how and by whom?
  8. One of the problems with translating results from one setting to another is that the system for radiologists training and who reads ultrasounds differs across settings. The authors state that gynecologists did the IOTA assessments and that (line 445) “experts” were “defined as gynaecologists with more than five years of ultrasound experience.” The authors should clarify this further—All gynecologists do some ultrasound. Therefore, does expert mean a gynecologist who has been in practice for at least 5 years? Or does it mean gynecologists who specialize in ultrasound where ultrasound represents a major (eg 50% ) of their practice.
  9. The sample size section is confusing, in part because the research question is not as clearly stated and the rest of the analysis doesn’t adhere to the primary research question. It appears the sample size calculation was done based on the population of women with inconclusive IOTA scores being the population of interest. I am assuming they are saying that 640 women were needed to expect 160 (25%) to have an inconclusive IOTA score. However, to have 80% power to detect a 5% difference between two proportions of 85% and 80%, at a confidence level of 95%, 901 women are needed, so it is unclear how they arrived at 640. This needs to be clarified.
  10. Table 1 should include percentages with all numbers, not just some of them.
  11. It is notable that 31% of women with inconclusive IOTA scores had malignancy. This seems very high and obviously implies the IOTA score misses a lot of malignancy. The authors should comment on how this compares with previous reports.
  12. Line 227: Overall, IOTA was more accurate than ROMA or RMI in diagnosing a 227 malignant mass (94% vs 84% and 89%, p=<0.001) (Table 3). The authors should state how “accuracy” is defined?
  13. It should be better described who determined the IOTA score and RMI score as both of these include subjective variables related to ultrasound features. Was it a single individual who did all of them or a single person at each site or a small group of people who did the scoring for all sites? If there was uncertainty or disagreement how was this resolved?
  14. A table showing the actual equation/variables that comprise IOTA simple rules, RMI and ROMA would be helpful as many readers will not remember the components/equation.
  15. The information in 3.6. Performance in cancer centre vs general hospitals (for patients with ovarian pathology) seems to indicate that the advantage conferred by expert ultrasound review was limited to the cancer center, which is not surprising. However, this finding is tempered by the fact that the prevalence of malignancy in the cancer centers was very high which would make it more likely that radiologists would call malignancy. This is alluded to in lines 386-7, where the authors acknowledge that IOTA had higher sensitivity in the cancer center. The sensitivity of a test should not depend on prevalence, only the PPV and NPV. The fact that different sensitivities were observed implies that the test does not function the same way in the cancer center vs the community hospital—it is not the “same” test. The authors should comment on this in the Discussion.
  16. Table 8 is appreciated as many studies do not report performance for detection of early stage disease which is obviously an important clinical goal. It is interesting that the effect of expert ultrasound review is more pronounced in this group which is not surprising given the relatively lower sensitivity of tumor markers in early stage disease. The histologies of the early stage cancers detected however should be shown. It would also be interesting to know whether the performance of the different strategies varied by type 1 and type 2 tumors.
  17. The authors should comment on the fact that since all the patients were already scheduled for surgery, they represent a selected group of people. In clinical practice, the main clinical dilemma is not whether to send a patient to GYNONC or not, but whether a patient needs surgery at all. The authors should acknowledge that their results do not show these algorithms are useful in informing the decision for surgery.
  18. Line 405: The authors state that given the practical realities of the cancer center and community hospital, the more specific test may be preferred in the cancer center vs the more sensitive test in the community hospital. While this is an interesting discussion and acknowledges that these algorithms do not operate in a vacuum divorced from other clinical factors, the authors seem to assume that all women referred to GYNONC get laparotomy. While this may be true in their cancer center this is far from true overall. In the US, the overwhelming majority of cases done by GYNONCs on an apparent isolated adnexal mass are done laparoscopically. If the authors wish to keep this argument they should acknowledge that it only applies to their cancer center and is not necessarily generalizable.
  19. Line 434 “In this study, assessments by IOTA 434 were done by gynaecologists who are not ultrasound experts” should be in methods.
  20. Line 443 “Internationally published training requirements for an expert ultrasound examiner specified that the expert should spend most of the time performing ultrasound examinations “
  21. I am not aware of any such internationally agreed upon training requirements. This needs a reference.

This study, like other studies that evaluate these algorithms, conclude that a particular algorithm is superior in differentiating ovarian cancer. However, all these studies fail to account for other relevant clinical factors that drive management, many of which make these algorithms irrelevant from a practical standpoint. For example, if a woman has a CT scan positive or suspicious for carcinomatosis, she is going to have surgery by a gynecologic oncologist (or a biopsy and neoadjuvant chemo). It doesn’t matter what the IOTA score is or the RMI or ROMA or any combination of the above. Similarly, CA125 is widely used and routinely evaluated in postmenopausal women with adnexal masses. A postmenopausal woman with a mass and a markedly elevated ca125 is going to have surgery by a gynecologic oncologist, regardless of the RMI, ROMA or IOTA score. To really assess the clinical value of these algorithms, what is needed is a prospective randomized study that assess their performance in cases where obvious indications of malignancy such as ascites, evidence of metastatic disease, and elevated ca125 are absent, in which the management is dependent on the conclusion of the algorithm and then outcomes such as need for second surgery, morbidity and overall survival are assessed. In the absence of this, the discussion should acknowledge that conclusions of clinical efficacy are limited by the assumption that the result of these algorithms would actually drive clinical management, independent of other clinical factors.

Round 2

Reviewer 2 Report

The authors have responded to the main issues.

Additional minor changes are suggested:

Line 66: Suggest changing “leading” to “potentially leading” as data to show actual worsened prognosis from rupture is weak at best.

Line 82. Would not refer to HE4 as a “new marker” as it has been around for many years. Suggest changing the sentence to read “Among other markers, HE4 has been the most promising.”

Line 92. Would change “is” to “are” as it refers to ultrasound characteristics.

Line 97. Would change “Patients who could not be classified by…” to “Patients whose masses could not be classified by…”

Line 135. Would change the way secondary question 3 is worded as it is currently not worded as a question. Suggest “Does the performance of these tests vary by menopausal status, setting (cancer specialist centre vs general hospital) or tumor histology or stage?”

Table 1: would show the equation for ROMA, as this is shown for RMI

Line 190. Change to “Blood was taken..” to keep consistent tenses.

Line 196. Suggest changing “gynaecologist’s” to “surgeon’s”

Table 2. It appears that the percentages apply to the row for row 1 but then apply to the column for the remainder of the table. Please clarify in the table footnotes if these are column or row percentages

Lines 384-387. This is somewhat confusing as to which tumors were in this group. Was this limited to only those not associated with elevated ca125? While normal ca125 is more common among non-epithelial tumors, a slightly elevated ca125 will still be observed particularly if there is advanced disease. The authors should clarify if they assessed 28 non-epithelial tumors not associated with elevated ca125 or all non-epithelial tumors. Although not shown, I am assuming the confidence intervals around the sensitivity/specific overlap with those of epithelial tumors hence the comment about lack of statistical significance. This seems to be likely due to a lack of power since the number of non-epithelial tumors is limited. If so, a statement such as “This analysis was limited by the relative rarity of non-epithelial tumor” would be appropriate.

Line 428. Would change to “Although a test’s sensitivity is not expected to vary…”
